# Proposed Management System and Response Estimation Algorithm for Motorway Incidents

Sotirios Kontogiannis  and Christodoulos Asiminidis *

Laboratory Team of Distributed Microcomputer Systems, Department of Mathematics, University of Ioannina, 45110 Ioannina, Greece; skontog@cc.uoi.gr
* Correspondence: chasiminidis@cs.uoi.gr; Tel.: +30-26510-08252

**Abstract:** Motorway's personnel tasks management and incidents monitoring, and response are critical processes that contribute to the motorway's orderly and smooth operation. Existing management practices utilize SCADA technologies that control motorway actuator systems as well as various means of personnel communications mobile technologies. Nevertheless, contemporary motorways lack a unified incident response solution that tracks resources, sends notification alerts when necessary, and automates incident resolution. This paper presents a new holistic and unified management and response system called Resources Management System (RMS). This system was originally implemented as a generic motorways resources management system for the EGNATIA ODOS motorway that uses it in Greece. The implemented RMS provides real-time resources tracking, personnel utilization algorithms, and data mining capabilities towards incident confrontation. It operates as an incidents' collection and resources central communication interface. It is also capable of incident response and completion time categorization; real-time tunnel exits sensory monitoring, staff mobilization, and tracking system. Furthermore, the RMS includes machine learning methodologies and smart agents (bots) for solving the problem of estimating and evaluating the response and completion time of incidents based on previous successful incidents' confrontations.

**Keywords:** distributed sensory systems; Resources Management Systems; incident response systems; location management; IoT; data mining; knowledge mining; machine learning; smart algorithms

---





## 1. Introduction

With the development of the Internet and cloud technologies, machine learning has come to play an essential role in our daily lives. In recent years, a microprocessor can enact any task requiring far more than only a specific number of instructions or process data implementing models and classification rules that need to be strictly algorithmically followed. Contemporary computers and cloud services processing capabilities have led to the implementation of sophisticated artificial intelligence algorithms capable of unsupervised learning. The appropriate algorithms define resource tasks and responses based on the incidents. Additionally, the machine learning processes can be easily implemented for data mining and model training to predict and test the system responses [1].

Road transportations occupy the largest share in freight and passenger traffic among the other categories of transportation. Safety and management of operations are two of the most critical factors to achieve further growth and quality of service on the motorway. To achieve this, incident tracking and intelligent automatic suggestions are the main prerequisites [2].

Road tunnels safety is a critical factor as tunnels are significant parts of the motorways' infrastructure. Technological improvements over the last two decades offered substantial improvements to tunnel safety [3]. Nevertheless, the increased use of the motorways and therefore road tunnels lead to road accidents increase. Resource management systems are the critical steps towards automation and safety as they offer the appropriate response protocols and resources management recommendations to the motorway operators [4,5]. Such

systems use supervised learning algorithms, deep learning, smart-bots, and automated suggestions via predefined assisted learning incidents planning towards industry 4.0 and autonomous systems.

This paper introduces a new holistic Resources Management System (RMS) for emergencies and processes management in motorways. It provides real-time resource mobilization and tracking. It operates as an incident response system by collecting incident emergencies using its resources central communication interface. It provides an incidents evaluation process by categorizing responses using smart algorithms and bot responses. It also provides interfaces and dashboards for monitoring sensors and actuators along the motorway in real-time.

The proposed holistic RMS management processes (incidents response and resources management) are automated through a machine learning methodology. The proposed system emergencies include motorway events and automated response for such events. The proposed RMS, along with the resources management, incidents evaluation-estimation algorithms, and sensory interfaces, can achieve the following key objectives: (a) development of a holistic technological system for managing motorway human resources daily operations and equipment tracking; (b) data collection of sensory information (specifically implemented for tunnels smart exists) [6,7] acquired by operators of motorway traffic control centers; and (c) use of intelligent communications system and appropriate supervised and unsupervised machine learning algorithms to effectively deal with emergencies.

## 2. Related Work

In this section, related work of technological systems used for notification endpoints and incidents is presented. Most of these system's capabilities can be used as part of a technological motorway management system.

An application proposed by [8] called smart disaster notification system alerts people before a natural disaster and provides information about the nearest shelter's shortest path. This application receives the data about the weather condition based on the user's locations. Therefore, the system estimates the probability of the disaster and sends back a smart disaster notification based on weather data received. Several companies offer SCADA [9,10] as a concentration technological means of motorways' equipment monitoring and controlling. Additionally, they notify the motorway users of potential risks of arbitrary information applicable locally via Traffic Management System (TMS), telemetry messages, or traffic light signs like [11,12]. The SCADA system also has a significant appliance in road tunnels, such as lane circulation obstructions due to obstacles, motorway staff maintenance tasks, or even traffic divergence to side streets due to critical motorway incidents.

The most popular search engines for disaster safety monitoring and reporting to the public are studied by [13]. Motorway human resources can also use such mass notification capability. Results have shown that the first search engine is Google, followed by Yahoo and Bing. Related links also appeared for Android and iOS. An extensive study conducted by [14] incident management and monitoring application that tracks personnel sends and receives notifications.

A critical infrastructure system designed for building fire emergencies called Emergency Adaptive Routing (EAR) is proposed by [15]. The work is based on end-to-end real-time communication, which preserves the desired delay in message propagation and power-level adaptation. The architecture adapts hazard application features such as fire expanding, shrinking, and diminishing. The EAR mechanism achieves very good real-time packet delivery compromised to fire emergency.

The emergency alerts on campus for students' safety in educational institutions across Canada are introduced by [16]. Even though this work has made an enormous breakthrough, the functional limitations of communications infrastructures and systems are still under consideration. The paper suggests that a Short Message Service (SMS) is the best solution to cover the need for an emergency incident, which contradicts an emergency incident, which contradicts the research proposed by [17].

An emergency notification system (ENS) for a natural disaster that interacts non-conventionally and is tailored to vulnerable groups of people has been proposed by [18]. Mainly, people that belong to vulnerable groups should subscribe to the system so that a suitable mobile device can be given, e.g., personal digital assistant based on the user's ability. Specifically, the simple emergency alerts fo(u)r all ontology built into a mobile phone application receives differentiated group alerts in emergency situations, including additional response information for the users with disability problems.

A crowdsource framework for a neighbor-assisted medical emergency system is proposed by [19]. The presented framework is called the Neighbor Assisted Medical Emergency System (NAMES). It reduces the ambulance response time to the emergency event. Patients can either call an emergency number or use the pendent/wrist belt, sending emergency alerts to the hospital clinic. Regarding incident response and resource planning solutions, the Call-Em-All company has developed an emergency notification system that uses XMPP protocols for mass texting, text to speech conversion capabilities, and voice broadcasting for up to 10,000 contacts instantly [20].

The SingleWire company has implemented an alert software for mass notifications called Informacast. It uses SMS, e-mail, calls, and custom application dialog(s), custom mobile phone applications for organizations and companies' facilities management, as well as panic buttons and IoT integration [21].

The Fireeye company has released MIR 2.0, a web-based incident response and a highly configurable system with indicators of compromise (IOC). IOC scanning and incident analysis, dissemination, and management use a series of wizard steps. This system offers voice call and e-mail capabilities towards organization groups [22].

Tekmon company implemented a secure web-based notification system called Tactical, with an automated IOC configuration that uses SMS, voice calls, and e-mail capabilities and appropriate escalation logic. Tactical advanced escalation logic ensures that all recipients will gradually receive the alert by changing its transmission technologies if the alert is not received [23]. The Tekmon company also implemented a system that ensures immediate personalized notification and personnel activities monitoring (tasks) called Daily operations [24].

Strasser et al. [25] studied the problem of computing rapidly the point-to-point shortest paths in massive road networks to predict traffic. They propose an algorithm called Customizable Approximated Time-dependent Contraction Hierarchies through Unpacking (CATCHUp) through Unpacking. It combines indexing the road network, the traffic patterns where the traffic patterns are used as an input of the query step, the shortest paths using the Dijkstra's algorithm, and the computation of the best suboptimal paths. In this work, streets are modeled as arcs, street intersections are modeled as nodes, and travel times are modeled as scalar arc wrights. They focus on space consumption and running times. They conducted experiments considering four road networks, each consisting of a different number of nodes and arcs. Experimental results have shown in terms of preprocessing, which considers time and query times that the CATCHUp algorithm outperformed related algorithms. Their approach offers competitive query times while keeping memory consumption on reasonable levels. By far, the algorithm is the first to achieve small index size fast and exact queries. Finally, the algorithm is the fastest preprocessing, competitive query running times and up to 38 times smaller possible indexes than other state-of-the-art techniques.

## 3. Proposed Resources Management System Architecture

The authors of this work propose a new resources management system for motorways and road tunnels. Following a holistic approach that collects and evaluates resource information, the system attempts to integrate both notification capabilities, incident response, and facility management of operations and equipment. The RMS can monitor field personnel, sending notifications to personnel on the motorway using the appropriate mobile phone application. It receives data from the controllers installed on the motorway tunnels.

The IoT data include measurements such as temperature, humidity, $CO_2$ ppm concentration, barometric pressure, and the detection/speed of vehicles and/or moving people close to the tunnel exits. Controllers provide such information that includes Bluetooth sensors in scanning mode. These controllers trace vehicles and people inside the tunnels. These controllers are placed at the motorway tunnel exits.

The RMS services fulfill the RMS framework's three-management axis, offering a unified management functionality that includes the following: (1) human resources real-time monitoring and evaluation-recording of their response, completion time, and actions; (2) incidents forwarding and response; (3) equipment and controllers/actuators' measurements acquisition in a transparent-homogeneous way. In-depth, the proposed RMS can offer the following services:

- Authentication service for user identification for logging into the RMS using the web platform or the appropriate Android App used by the human resources, which are separated into teams based on their duties on the motorway.
- Facility management capabilities tracking service with applicable encryption for acquiring resources GPS location in real-time. Utilizing the proposed by the authors' distance algorithm and completion time to select the closest personnel to respond to each incident. Applicable response time algorithm for the process of personnel evaluation towards incidents response.
- Facility management capabilities with the submission of tasks as notifications and the ability to use the tasks completion time for the process of evaluating personnel tasks responses, predicting their future responses, and automatically assigning tasks based on the response via the RMS.
- Incident response capabilities via alerts service and incident recording service for sending alerts from the headquarters to the road patrol personnel through the mobile phone application and vice versa. Applicable distance algorithm for the real-time and transparent to the operator selection of the appropriate field personnel to intervene in an incident. Classification of incidents and the ability to predict future incident responses using incidents response algorithm and assign mitigating personnel accordingly.
- Equipment management capabilities by the equipment measurements service. This service overs a unified protocol for collecting real-time measurements, taken by the personnel on the spot sensory-equipped controllers or the Bluetooth sensory concentrators placed inside the motorway tunnels. Such information is directly visualized to the RMS central point of control (motorway headquarters) via appropriate RMS dashboards per controller-concentrator. Furthermore, to integrate sensory measurements to an alert message, the RMS mobile phone application can also be used. The sensory measurements utilize the CoAP protocol and the UDP/nCoAP Java library [26] to send CoAP resource observations to the RMS central point of control [27]. Proposed RMS system functionality for a holistic approach system capabilities and existing systems such as Tekmon daily operations [24] and MIR 2.0 [22] is presented in Table 1.

The authentication service is implemented for users logging onto the RMS. The HTTP service is used over SSL. The personnel real-time tracking service is responsible for sending motorway field personnel location information to the RMS central database. This service is implemented over a lightweight Node JS HTTP service using the GET method over the 9090/TCP port where the ID user, the user group, and the coordinates are periodically sent, using JSON notation from the user's location to the RMS database cache. These data are stored in the RMS database and presented to the central office RMS operator using motorway GIS real-time layered map representation similar to the one mentioned by [5] and [28]. Furthermore, the coordinates are encrypted using the AES-128 algorithm to increase real-time protocol security.

**Table 1.** RMS services functionality over existing facility management and incident response systems.

| Holistic System Capabilities | RMS | MIR | Daily Operations |
|---|---|---|---|
| Incident Response (alerts) | ✓ | ✓ | ✓ |
| Facility Management (forms-notifications) | No custom Tasks Manager | ✓ | ✓ |
| Facility Management with the use of sensors (GPS/NFC/Bluetooth/other) | GPS/BLE | GPS/other | NFC |
| Smart Algorithms for Incident response | Distance Algorithm, Response time Algorithm | Fixed Assessment Plan | Fixed assessment Plan/Mass notification Tactical interface |
| Smart Algorithms for Facility Management | Completion Time Algorithm | System Reports | System Reports |
| Sensory Measurements Collection | ✓ | ✓ | No |
| Mass notification Capabilities | No | Yes/Escalation logic | Yes/Escalation logic |

The notifications/alerts service is responsible for sending notifications in case of an incident in real-time from the RMS web manager app to the field personnel deployed on the motorway through the RMS mobile phone application (Android). This service is implemented using Node JS, and a lightweight HTTP application service at TCP/9091 port, where the personnel periodically query (every 30 s) using HTTP PUT notify requests using a mobile application service implemented in the RMS Android App. This service can also send HTTP POST requests of JSON annotated notifications to the RMS central database. These notifications are visualized in real-time to the RMS web manager dashboard. The architecture of the RMS is shown in Figure 1.

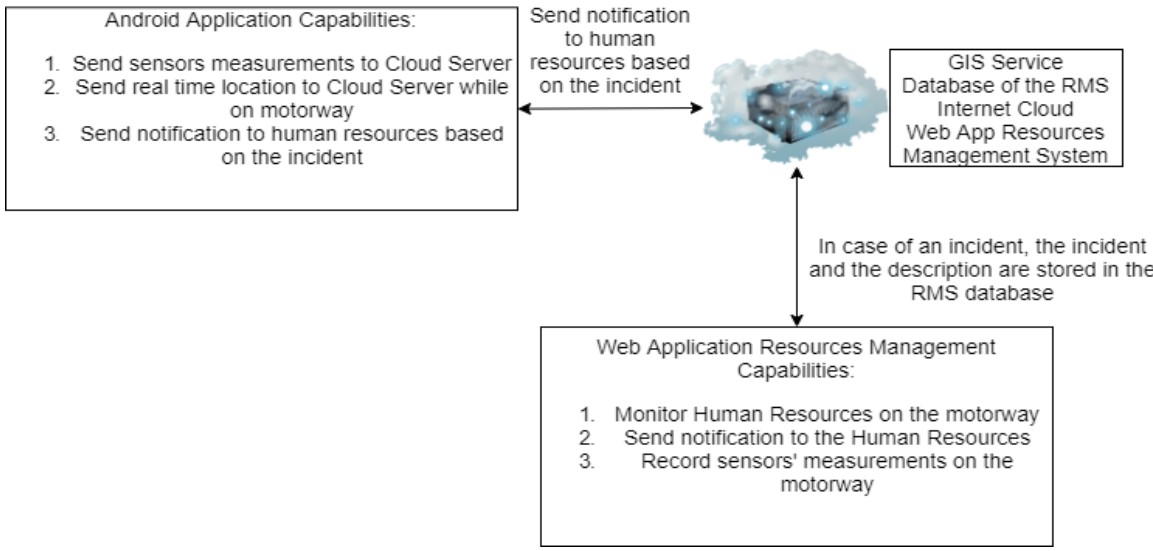

**Figure 1.** Proposed resource management system architecture.

All notifications or tasks are collected to the RMS web panel that includes: a dashboard map for monitoring personnel (GPS tracking service), notifications (tasks)/alerts dashboard monitoring status of existing notifications or alerts, notifications (tasks) and alerts manager and the equipment measurements dashboard that monitors in real-time controllers' sensory measurements.

The equipment measurements service can receive measurements from Bluetooth sensory controllers placed in tunnels, or other sensory equipment, directly to the RMS

database. For the process of direct database collection of RMS measurements, each controller is connected to the Internet and communicates with the database service over a private VPN network. Measurement's collection intervals vary from 1–5 min depending on the type of sensory equipment and type of measurements. Appropriate dashboards called sensory dashboards are included at the central RMS web interface to interact with each controller, monitor sensory measurements, and even provide per controller actuating output such as sound alerts, led stripes on/off. Such a VPN network of interconnected actuators and concentrators communicating with the central database service in real-time is part of the RMS distributed system. RMS distributed system capabilities and scalability, using a single form of concentrator communication using a central VPN service and direct database connectivity, signify the proposed RMS system's holistic approach towards sensory equipment.

Controller measurements can also be received by the patrol staff who are near the controllers via Bluetooth through the RMS mobile phone application. In such cases, the measurements service uses the UDP protocol, and the measurements are sent using the CoAP PUT method. The implementation uses the Constrained Application Protocol (CoAP). CoAP main functionality is to immediately send messages from the mobile phone application to the RMS. That service is not bidirectional so that the Android App users who are on the motorway can update measurements and acknowledge their updates to the RMS but cannot receive previously updated measurements [29].

## 4. Proposed Resources Management System Smart Capabilities

The RMS system implementation includes two new machine learning algorithms apart from the proposed distance algorithm. These algorithms are used from its existing RMS bots or proposition interfaces to process automated selection of personnel towards incidents or tasks. The algorithms used are the response and completion time algorithms. The purpose of these algorithms is to evaluate human resources towards incident response and task completion time accordingly. RMS supports incidents dissemination, sending alerts or notifications to the appropriate personnel in real-time for incidents on the motorway, and tasks that need to be completed. The tasks and notifications are recorded based on the team personnel. Furthermore, RMS monitors personnel, by visualizing their GPS location on a map in real-time and notifies them in case of an incident. For assigning notifications per recorded incident into its motorway field teams, RMS uses the teams' real-time tracked position and bot, which automatically decides which group is to dispatch based on a basis-distance algorithm from the incident as well as the team's specialization and ranking.

Motorway personnel who are patrolling on the motorway can use the RMS mobile phone application. The application tracks the personnel's GPS coordinates and monitors their placement in real-time using an appropriate dashboard map. RMS mobile phone application users can record an incident or notice the incident that needs to be mentioned to the motorway central. They can record measurements that are taken from nearby placed sensors. They can take screen captures and upload them as indicators to the RMS database and initiate an incident.

The RMS categorizes incidents based on the category of the incident, incident description, and notifications sending time from RMS to resources on the motorway, personnel arrival time at the incident, and incident completion time. The main RMS incident categories are: category 1—accident; category 2—immobilized vehicle/obstacle; category 3—pedestrians; category 4—animals; category 5—opposite direction; category 6—fire; category 7—clearance; category 8—traffic conjunction; category 9—bad weather conditions; category 10—fatal accident; category 11—accident with injuries; category 12—property damage accident; category 13—fire in a vehicle; category 14—traffic events; category 15—motivation/conquest; category 16—intense weather conditions; category 17—police incident; category 18—other.

The RMS is separated into 6 human resources teams as follows:

1. Application Manager (AP) is responsible for controlling and monitoring the rights access of the resources management system.
2. Tasks Force Team (TFT) which is responsible for controlling and monitoring the human resources on the motorway. This team is also responsible for alerting human resources personnel by sending them a notification in case of an incident/task that occurs on the motorway.
3. Technical Support for Electronics and Automation Systems Team (TSEAST) which is responsible for the maintenance of electronics and automation systems.
4. Technical Support for Electronic and Mechanical Systems Team (TSEMST) which is responsible for the support of electronics and mechanical systems.
5. Electromechanical Maintenance Team (EMT) is responsible for the maintenance of the electromechanical part.
6. Motorway Incidents Team (MIT) is responsible for the incidents that occur on the motorway. Their primary duty is to address motorway incidents. They can also be assigned tasks.

The RMS can monitor the following personnel actions in the motorway: Personnel incident arrival, incident completion, submission of notification reports, and incident images. Using the mobile application, personnel can receive notifications in real-time from the motorway operators, send notifications back to the operators, and images from the incident area. The resources management system is shown in Figure 2a, where the user registered into the RMS is an operator that belongs to the TFT. In this case, as shown, all team personnel has been selected to be tracked and monitored on the map. In this example, only one is shown on the map which belongs to MIT (Motorway Incidents Team). The android application shows that the user is logged in, displayed with the green circle as illustrated in Figure 2b. The position error is displayed next to the orange circle and the MIT's position on the motorway is shown on the map.

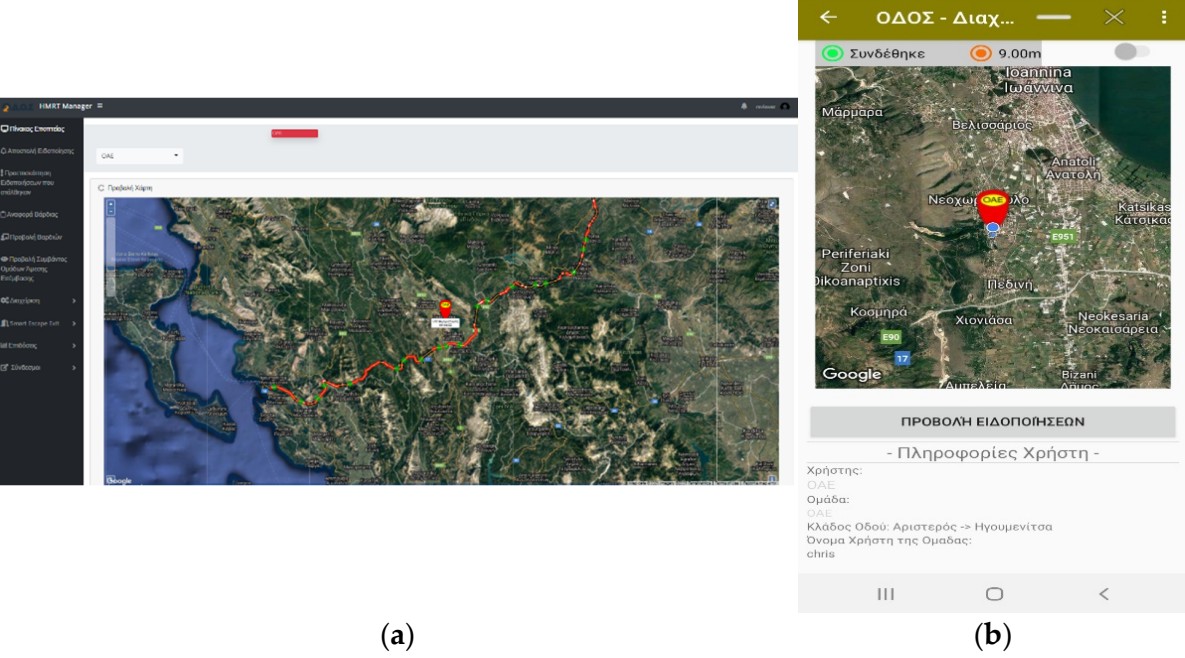

(**a**)                                         (**b**)

**Figure 2.** The resources management system consists of (**a**) information system; (**b**) Android application.

The RMS system smart algorithms are using the following metrics:

- Starting time of an incident: Time that automatically is inserted into the database when a user of RMS sends a notification to a personnel team for an incident or task.

- Notification confirmation of reception: Time that the personnel team on the motorway received their assigned notification on their smartphone RMS application.
- Incident response time: Time automatically inserted into the database when a personnel team is tracked close to the incident. The distance between the incident and the group is upper bounded and less than a threshold (km). This distance is set to be the length of the largest motorway's tunnel, which is 4 km. That is because smartphones do not receive GPS coordinates precisely and accurately when human resources are in tunnels. The personnel fixed location must remain unchanged to the one received before their vehicle enters the tunnel (the Android App sends the exact coordinates to the RMS every 16 s). The biggest distance that a personnel team can be considered from the RMS that can respond to an incident is given by Equation (1).

$$maxd_i = \max(tunnel_{ten}) + \frac{2}{3}\max\left(\frac{dS}{dt}\right)T_{GPS} \tag{1}$$

where max is the length of the largest tunnel of EGNATIA ODOS motorway, $tunnel_{ten}$ is the tunnel or motorway speed limit, and $T_{GPS}$ is the transmission period of field team GPS location to the RMS tracking service. Focusing on the case study motorway EGNATIA ODOS, the largest tunnel has a length of 4.6 km = 4600 m, the speed limit is 130 km/h = 36.1 m/s and that means that the biggest distance is considered from the location of the incident to the last received GPS coordinate of the field personnel which in this case is 4985 km.

- Task Completion time: That is the time of task completion and lane restoration as indicated by personnel via the RMS mobile application.
- Teams' response ranking: It is calculated as each team's mean incidents response time overall teams mean response time in a specific motorway section, as calculated for all incident categories based on Equation (2) and is measured in minutes.

$$R_i = \frac{\bar{r}_i(t)^2}{\sum_{i=1}^{n} \bar{r}_i(t)} \frac{60}{u_{limit}} \tag{2}$$

- Teams' completion time index: It is calculated as the normalized mean tasks completion time per category over all other teams' completion time for the same incident category. It is an index value between 0 and 1. If it is close to 1 that indicates a fast completion time for that category. It is expressed by Equation (3).

$$C_{ind_i} \begin{cases} 0 \,, \, c_i(t) \leq 0.01 \\ log2\left(\frac{c_i(t)}{\sum_{i=1}^{n} c_i(t)}\right), \, 1 \geq c_i(t) > 0.01 \end{cases} \tag{3}$$

The proposed RMS implements data mining over past metric values to estimate: 1. the motorway teams that are closer to the incident (this is mainly calculated from the distance algorithm). 2. Teams that are closer to the incident that has previously been rated according to its incident category, based on the lowest (best) response time and team's motorway placement. 3. Teams that have completed incident actions or management tasks. That is achieved using the completion time algorithm. The motorway operator can select either the distance or response or completion time to assign an incident or task to a team accordingly. The previously mentioned algorithms (1, 2, 3) are presented in the following subsections.

### 4.1. RMS Distance Algorithm

The authors propose an automated way of selecting the personnel closer to the incident location based on the lane direction, distance from a specified 0 km point in the motorway, and the incident response time metric expressed in Equation (2) acting as a weight factor. This algorithm is called the distance algorithm, and its flow diagram is presented in Figure 3.

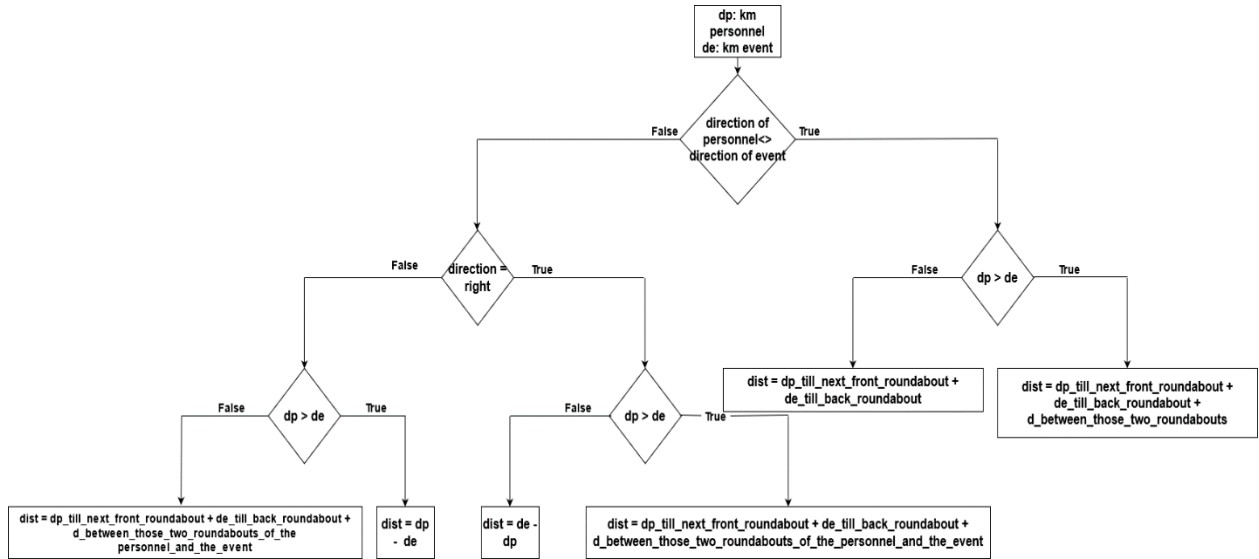

**Figure 3.** RMS distance algorithm flowchart and logic.

The steps for calculating the distance between an incident occurrence either in a tunnel or open road and the field personnel teams through the Android App tracking service that has been implemented in the RMS is as follows:

Step 1: The system linearly interpolates the latitude and longitude coordinates of the users who are connected to the Android App, the roundabouts of the motorway, and the coordinates of the event based on the EGNATIA ODOS motorway mileage in km. Considering that mileage starts at zero km at Igoumenitsa, city, West Greece, and ends at Alexandroupoli Kipi, a town in East Greece at the 658th km, the mileage is the same for both directions, left and right of the motorway. The mileage is an increasing number in the left direction of the motorway where it starts in Igoumenitsa and finishes in Kipi and a decreasing number that starts from Kipi and ends at Igoumenitsa.

Step 2: The system checks whether the direction of the personnel and the event are in the same direction on the motorway. If both are in the same direction, it proceeds to steps 3a, 4 else to steps 3b, 4.

Step 3a: If the personnel are in the same direction (team, event), then the system checks if the linearly interpolated distance of the team on the motorway is greater than the linearly interpolated distance of the incident:

    i. If $d_{team} \leq d_{evt}$, distance equals the distance between the distance of the team and the event.

    ii. If $d_{team} > d_{evt}$, distance equals the distance between the distance of the team and the next front roundabout plus the distance of the event and the back roundabout plus the distance between those two roundabouts.

Step 3b: If (team, event) are in opposite directions then the system calculates the linearly interpolated distance of the team on the motorway, the distance of the next roundabout, the incident's distance, and the previous from the incident roundabout distance. It then calculates the distance difference of the team and its next roundabout $d_{tn}$ and the distance difference between the incident and the team's next roundabout $d_{in}$. The sum of these two distances is the (team, incident) distance.

Step 4: Multiply the calculated team distances from the incident with the distance metric from Equation (2). Select the team with the minimum normalized distance.

### 4.2. Proposed RMS Response Time Algorithm

For the process of response time calculation, the response time algorithm is introduced. The estimated response time of a personnel team (in min) per km on the motorway. The response time of every team is calculated using the stored data in the RMS database and a mean estimation of the optimal response time for every team per location, based on the team's km real-time placement on the motorway. The authors propose the RMS response time algorithm to evaluate group incidents response time over different parts of the motorway. The purpose of this algorithm is to estimate teams of motorway personnel based on their response time at an occurred incident at a specific motorway location.

The functionality of the RMS response time algorithm is perceived through an agent (smart bot), which is implemented on the RMS platform and can extract and evaluate past team data from the RMS database. After filtering, these data are the training data for the response and completion time algorithms (a similar approach is also proposed by [1]). The output of the response algorithm is the team estimated mean response per incident concerning the motorway absolute motorway position as measured in km (left or right lane accordingly).

The output of the completion time algorithm is the estimated mean completion time of the personnel per-incident and incident description category depending on the lane on the motorway. The functionality of the Intelligent Agent and the machine learning algorithms are illustrated in Figure 4.

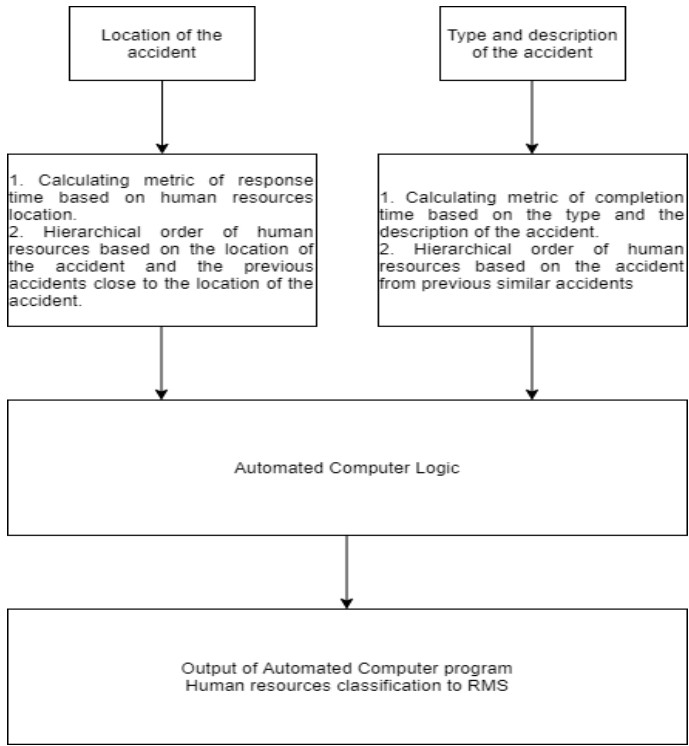

**Figure 4.** Illustration of the RMS functionality of machine learning algorithms—metrics calculation and intelligent agents.

The Intelligent Agent—smart bot presents to the user of the RMS web interface the values of metrics 1 and 2 of an incident that are inserted into the database as well as a list of the available personnel teams based on the response time algorithm and the mean response/completion time concerning for the index of every group close to the incident. The first group that appears on the data table is the best optimal suggestion of the smart bot for the specific incident. The difference in groups' response time and the difference in completion time are displayed through the appropriate web interface provided by the RMS web manager.

The authors propose the response time AI algorithm. This algorithm aims to estimate each motorway group's response time per incident on specific motorway locations. This is achieved using data mining over past incidents. The data mining process is based on the data coming from the RMS database. The RMS coordinator sends a notification, and subsequently, the team of field personnel that is close to the incident on the motorway receives the notification on their RMS mobile phone application. The distance between the incident and the team is calculated using the Haversine formula. Then, the team user notifies the RMS web service that has responded to that incident and, thus, that the incident is under its supervision. The functionality of the response time algorithm using the Intelligent Agent (smart bot) consists of the following steps:

1. Periodically receiving data from the RMS database. These data are part of the training data set.
2. Separating data based on the direction of the personnel that is on the motorway. There are two directions, the left and the right one. From now on, every step described is being explicitly applied separately for each direction.
3. Filtering input data. Choosing from the training data set those data that there is response time metric field. A process of cleaning is followed repeatedly, as shown in Figure 5.
4. Data clustering process. The filtered data from step 3 are now clustered, applying the *k*-Means clustering algorithm using *k* = 2-clusters. Subsequently, the Sum of Squared Error (SSE) is calculated. If the value is greater than 1, we increase the number of clusters by 1 (k++). This process is being repeated once we get the value of SSE being less than 1 or the number of clusters greater than a threshold that we set, as stated in Equation (4).

$$2 \leq k \leq \sqrt{\frac{N}{2}} - 1 \qquad (4)$$

5. Once the value of SSE is less than 1, the numbers of centroids of *k* clusters that contain points in two-dimensional space (response time, km distance) are implemented using PHP-ML library [30] and R [31] at the RMS web service for the process of automatically training an intelligent Support Vector Regression (SVR) engine.

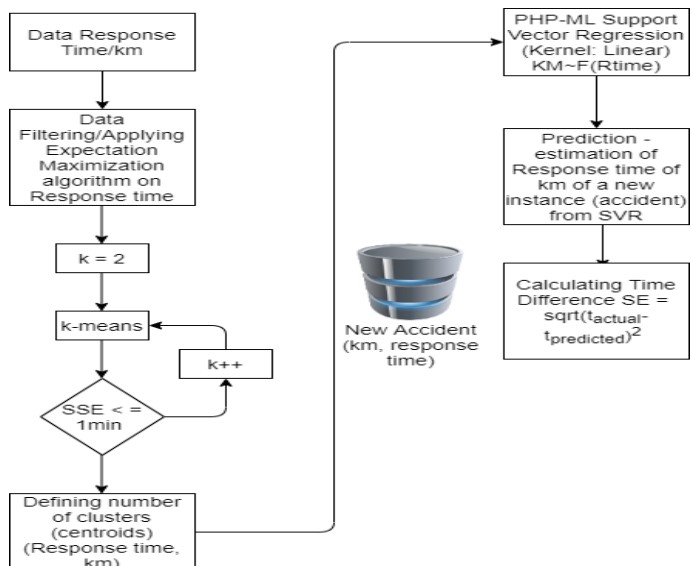

**Figure 5.** Illustration of the response time algorithm steps. Estimation of Response Time of motorway's field personnel when an incident occurs.

The SVR engine is applied to both motorway directions and gives the estimated response time in any location and direction that an incident occurs on the motorway. The

motorway's personnel are categorized into 6 different groups depending on the specialty and task. So, each group has been normalized based on the location of the incident, as shown in Equation (5):

$$NSE = \begin{cases} \frac{aSE + (1-a)\overline{SE}}{dS} \text{ ,} if\ SE > 0,\ \overline{SE} \neq 0 \\ \frac{(1-a)|SE| - a\overline{SE}}{dS} \text{ , } if\ SE < 0,\ \overline{SE} \neq 0 \\ \frac{\overline{SE}}{dS} \text{ , } if\ SE = 0 \end{cases} \tag{5}$$

where $SE = t_{actual} - t_{estimated}$ and $a = \frac{1}{3}$.

The parameter of the NSE is calculated in min/km. The last mean value of NSE is used as an evaluation parameter of the estimated response time for a personnel team. A multiple factor to the output of the choice of which team is chosen for the incident to act for the incident. For example, if there are two teams, the first is 20 km away from the incident, and its NSE value is equal to 1.5 min/km, and the second one is 35 km away from the incident, and the NSE value is equal to 0.7 min/km. Then, the results are as follows:

1.  First Group: 20 km × 1.5 min/km = 30 min.
2.  Second Group: 30 km × 0.7 min/km = 21 min.

Depending on the NSE value, the response time algorithm would lead the second one to the location of the incident, even though the first one is closer to the incident. Before applying the training data set to the response time algorithm, data should be filtered in case any outliers might affect the accuracy of the results of the algorithm. The response time algorithm flow is illustrated in Figure 6. In Figure 6, the data filtering process is depicted in dashed lines. The target of this process to find the response time clusters of teams.

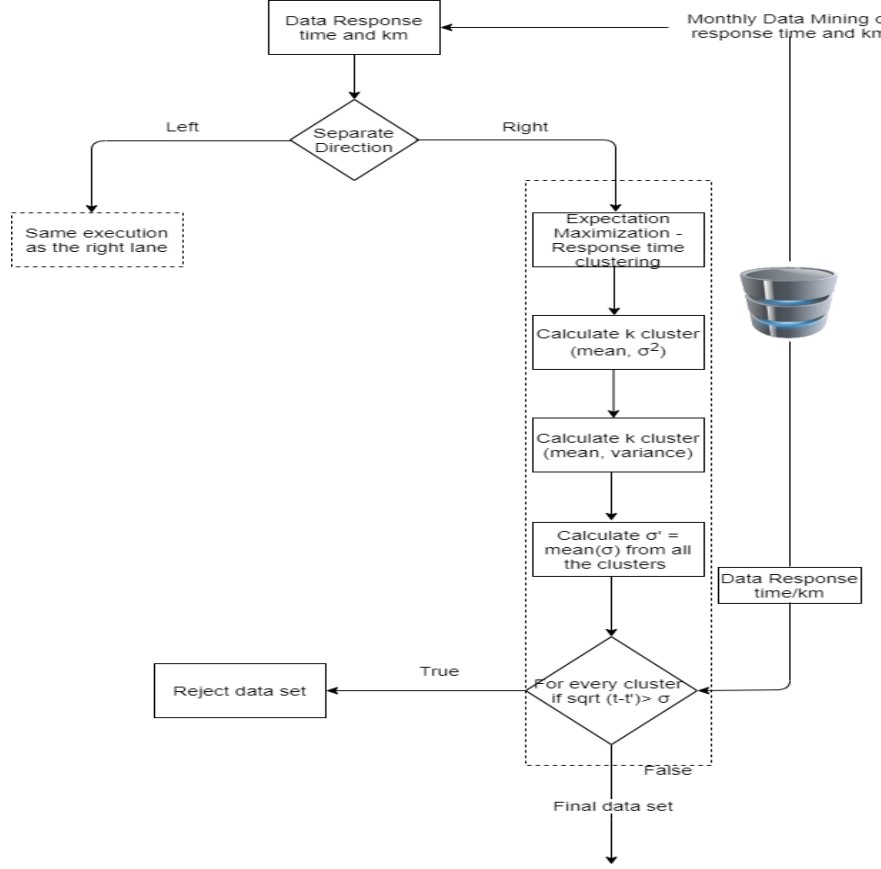

**Figure 6.** Flow chart of the response time algorithm data filtering and training.

The Expectation-Maximization (EM) is used on the response time of all the data sets in the first place under the condition that the response time is gathered follow the normal distribution.

The normal distributions of the data set as well as the calculation of the likelihood logarithm of the chosen distribution(s) as to the sum of all the probabilities of the data set points (A) extract are described in Equation (6):

$$L = log\frac{L(B)}{L(A)} = log(L(B)) - log(L(A)) = log\left(\prod_{i=1}^{N} p_{i_{dist}}\right) - log\left(\prod_{i=1}^{N} p_{i_{real}}\right) \qquad (6)$$

The result of $10^L$ shows the percentage % of fitting the data of the chosen distribution(s) to the real data. So, $L$ must be a negative number, s.t. $L < 0$, so that the training data set is acceptable to its data set. In any other case, the data set is rejected. After applying the EM algorithm, the mean value of all the standard deviations of the k-clusters from the EM algorithm and the so-called $\sigma'$ are calculated from the set of $\sum_{i=1}^{k} N_i(\mu_i, \sigma_i^2)$, as shown in Figure 6. Thus, Equation (7) is given:

$$\sigma' = \frac{1}{k}\sum_{i=1}^{k}\sigma_i \qquad (7)$$

All the data are filtered using the value of $\sigma'$ as the value of the standard deviation to the normal distribution they belong to (cluster they belong to or the difference between the value from the mean value of the cluster to be the minimum). The final filtering algorithm is described as follows:

S1:　For every point, calculate the cluster of EM that it belongs to. That is the mean value of the cluster as given in Equation (8):

$$\forall(t_i, p_i)find\mu_i\forall\mu_\iota \in \cup_N d_{istance} = min\{\mu_i, t_i\} \qquad (8)$$

S2:　Calculate the difference between the mean value of the chosen cluster i and every cluster from the set of cluster $k = 1, 2 \dots$ m as given in Equation (9):

$$\forall(t_i, p_i), v_i = (t_i - \mu_l)^2 \qquad (9)$$

S3:　If $v_i \leq \sigma'^2$, then the incident $i$ is acceptable as an incident of the training data set. In any other case, the incident is not acceptable and then, it is filtered.

### 4.3. Proposed RMS Completion Time Algorithm

For the process of task completion time calculations, the completion time algorithm has been introduced. That is the estimated completion time of motorways' personnel based on incident category task and task description. The completion time is determined using the data mining estimation and classification process on the data stored in the RMS database. In the RMS, the user sends a notification about an incident to online users and so the start time of the incident is inserted in the database. Each of the personnel team is closer to the incident, the RMS has stored the completion time used for the estimation of completion time of the incident, and the description to apply the second machine learning completion time algorithm. The purpose of this algorithm is to estimate incident completion time for motorway personnel based on incident category and description.

The algorithm's functionality is interpreted utilizing an agent (smart bot) deployed on the RMS web service and can extract task-related information from the RMS database, which is signified as the algorithm's training data set. The output of the algorithm is the estimated mean completion per incident category or task concerning the specific task description.

The completion time machine learning algorithm tries primarily to detect similar team tasks and then estimate the task completion time. The functionality of the algorithm using the Intelligent Agent (smart bot) consists of the following steps:

1.  Periodically post-processing data from the RMS database are part of the training set (several training sets of tasks descriptions or incident categories need to be maintained at the database—supervised mining process).
2.  Separating data based on the direction of the personnel that is on the motorway. There are two directions in total, the left and the right one. From now on, every step described is being explicitly applied separately for each direction.
3.  Filtering output data. Choosing the from the training data set those that there is completion time metric field and then a process of cleaning is followed repeatedly as shown in Figure 7.
4.  Data clustering process. The filtered data from step 3 are clustered using the *k*-Means algorithm setting primarily clusters *k* = 1. Subsequently, the Sum of Squared Error (SSE) is calculated. If the value is greater than 0.5, we increase the number of clusters by one (k++). This process is being repeated once the value of SSE is less than 0.5.
5.  Once the SSE is less than 0.5, the numbers of centroids of *k* clusters that contain points in two-dimensional space (completion time, category of incident) are implemented using PHP-ML library and R at the RMS for training the algorithm's k-Nearest Neighbors (k-NN) engine.
6.  The descriptions of the incidents have been classified based on the frequency of the incident in the RMS database. The Intelligent Agent (smart bot) displays all descriptions in descending order of the specific category of incident.

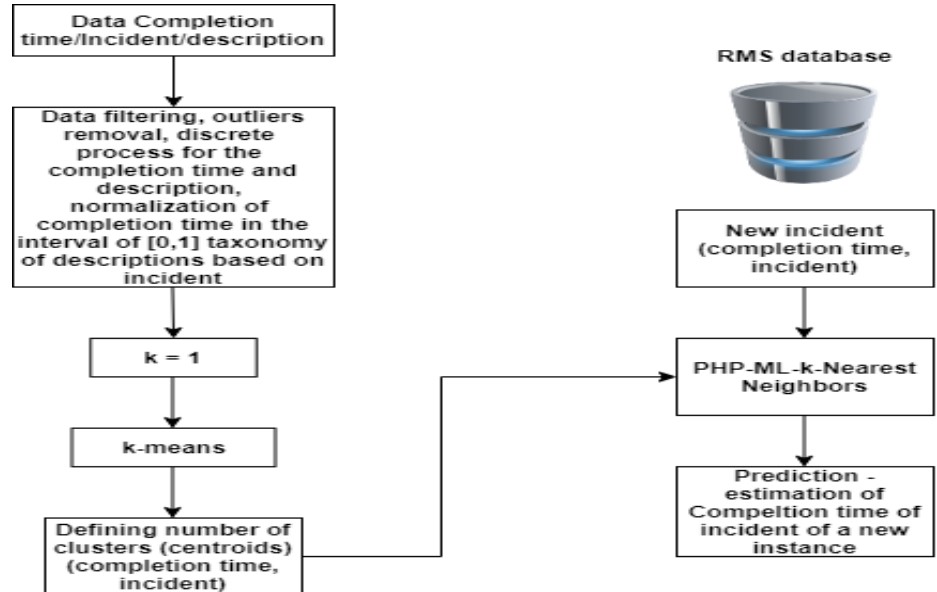

**Figure 7.** Illustration of the completion time steps. Estimation of Completion Time of motorway's field personnel when an incident occurs.

## 5. Proposed RMS Algorithms Validation

The previously proposed algorithms have been evaluated using the RMS system on the EGNATIA motorway. The system has been used for the duration of two months. The system application area is 69 km. During this time, the system's personnel included two operators and the seven personnel teams of the motorway incidents team (MIT). The incidents recorded using the RMS database during this time, incident response and completion times, and the accuracy of the proposed algorithms, are evaluated in the following sections. Recorded incidents data analysis follows.

Depending on the category of the incident that is given from the headquarters of the motorway and based on the statistical analysis from the annual recording of important incidents 2018, every incident is coded to a category of incidents described in section DATA. These data come from the pilot application of the RMS on a targeted part of the motorway, which starts at the length of 0 km (Igoumenitsa city, Eastern place of Greece) and ends at the 69th km (Ioannina city, Western place of Greece [2]). Most of the incidents that occurred in this section belong to category 2. That is why the training and evaluation of the response time machine learning algorithm per km of the motorway include incidents between 0–69 km of the EGNATIA ODOS motorway.

For the response algorithm training process, prediction, filtering, and automated data retrieval from the RMS database, the R programming language has been used. Regarding the statistical process of the RMS data, the data have been divided based on the direction. From the statistical process, we see that the completion time is no more than 300 min. Additionally, results show that the completion time to an incident is on average 30 min. 15% of the incidents require 30 to 55 min of mean completion time. The amounts of incidents that need more than 30 min to be completed belong to the second category. Finally, the number of incidents that are over 30 min belongs to the direction from Western to Eastern Greece, towards Igoumenitsa. Most of the incidents (51 out of 93) happen in the direction from Eastern to Western Greece, towards Ioannina, in which we have a variation of all the categories of the incidents. On the contrary, in the direction from Western to Eastern Greece, towards Igoumenitsa, categories 1, 2, 3, and 4 of the incidents appear and the total number of instances is 42 out of 93. Table 2 summarizes the total instances of the data set which are 93. 6 of them belong to category 1, 74 to category 2, 3 to category 3, 8 to category 4, 1 to category 5, and 7.

**Table 2.** The number of instances for each incident category.

| Category | No. of Instances |
|----------|------------------|
| 1 | 6 |
| 2 | 74 |
| 3 | 3 |
| 4 | 3 |
| 5 | 1 |
| 7 | 1 |

Table 3 shows the percentage of incidents on the motorway based on the km either in the left or right direction. It is shown that 32% of the incidents occur in both directions at the 18th and the 26th km. The highest percentage appears from the Western to Eastern side of the motorway, from Ioannina to Igoumenitsa. The 28% of the incidents happen between the 35th and 50th km evenly distributed to both directions.

**Table 3.** Percentage of incidents per direction per km.

| Percentage of Incidents | Direction | Km |
|-------------------------|-----------|-----|
| 32% | Both | 18 and 26 |
| 65% | From Western to Eastern | 0–69 |
| 28% | Both | 35 and 50 |

Table 4 shows that the completion time for left lane incidents. For the first incident category, the completion time is more than 50 min to be completed after removing the outliers. Without removing the outliers, it reaches up to 9 h. The completion time equals 30 min for the second category, whereas for the third incident category, the completion time is less than 5 min. The completion time for the fourth incident category is less than 25 min.

**Table 4.** Total completion time per category for the left lane.

| Category | Total Completion Time (min) |
|---|---|
| 1 | >50 |
| 2 | 30 |
| 3 | <5 |
| 4 | <25 |

Table 5 shows the completion time for right lane incidents. For the incidents that belong to the first category, the completion time is more than 100 min. Without removing the outliers from the dataset, the completion time reaches up to 6 h. For the ones that belong to the second category, the completion time required is 40 min. For the third one, the completion time is less than 5 min. For the fourth one, the completion time is less than 15 min. For the incidents that belong to the 5 and 9 categories, the completion time equals 5 min.

**Table 5.** Total completion time per category for the right lane.

| Category | Total Completion Time (min) |
|---|---|
| 1 | >100 |
| 2 | 40 |
| 3 | <5 |
| 4 | <15 |
| 5 | 5 |
| 9 | 5 |

The Completion time value data are normalized in the interval [0, 1] given by Equation (10).

$$norm_{ct} = total\_completion\_tim - \frac{\min(total\ completion\ time)}{\max(total\ completion\ time) - \min(total\ completion\ time)} \tag{10}$$

Moreover, the *k*-Means algorithm is used to train the model and predict new values inserting the model as described in the Proposed RMS completion time algorithm section.

*5.1. Response Time Algorithm Evaluation*

On the data set of the RMS as they have been inserted into the dataset and during the pilot application on the area of the 0–69 km of the motorway, from Ioannina to Igoumenitsa, the data filtering mechanism is applied. Initially, the data is separated according to the two directions, left (to Ioannina city) and right (to Igoumenitsa city). The first step of the data filtering mechanism includes the EM algorithm. The outputs of the EM algorithm are shown in Figure 8.

The results show there are two distinguishable clusters. Although one of them has missing data (red line), it is still included as normal distribution to the output of the algorithm. The red distribution is part of only 18% of the data set, while the rest of the data are represented as the green distribution. The red distribution $N$(3.38, 2.64) has parameters $\mu$, $\sigma^2$ that equal to 3.38, 2.64 and the green one $N'$(21.19, 11.06) has parameters $\mu'$, $\sigma'^2$ that equal to 21.19 and 11.06 accordingly.

The results of the right directions show that there are two clusters by which one of them has missing data (green line). Nonetheless, it is included as normal distribution to the output of the algorithm. The green line considered as normal distribution consists of only 15% of the whole data set whereas the rest are illustrated as the red line. The red line follows normal distribution $N$(15.37, 11.34) where $\mu$ = 15.37 and $\sigma^2$ = 11.34 as well as the green one $N'$(49.3, 4.17) where $\mu'$ = 49.3 and $\sigma'^2$ = 4.17.

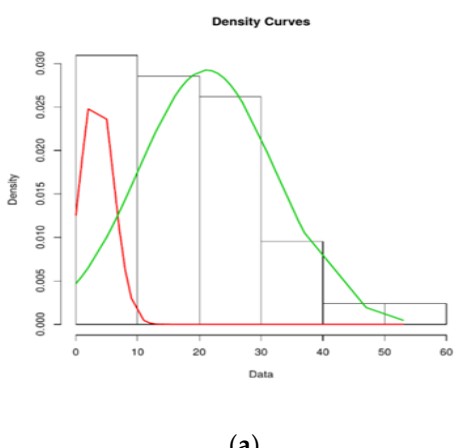

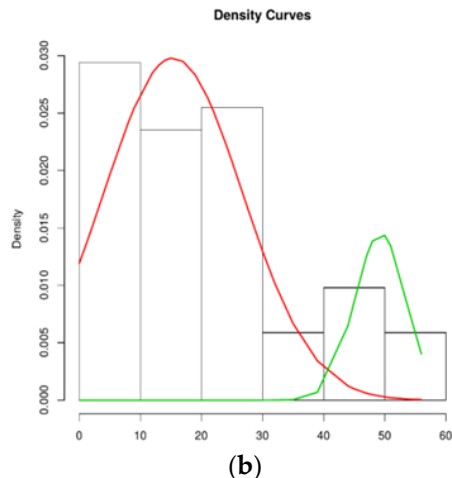

(**a**)                                                   (**b**)

**Figure 8.** The output of the EM algorithm for the (**a**) left direction towards Igoumenitsa; (**b**) right direction towards Ioannina.

Using the filtered data as training data for every direction, the k-Means algorithm is used with initial cluster number $k = 2$ until SSE < 1. The left direction, towards Igoumenitsa, creates 6 clusters where SSE = 0.934. The right direction, towards Ioannina, creates 4 clusters, wherein this case SSE = 0.565. Due to the non-normal distribution of response time per km while running the algorithm, because the data set is not clear and there are missing values, a margin process of clusters is proposed. That process is optional, and it contains the margin of clusters when the difference of their distance is less than 10 km. The merging of clusters follows the principle of weight and is described in Equation (11).

$$dt_c = \frac{1}{N} \sum_{i=1}^{t} w_i t_{C_i} \tag{11}$$

where $N$ is the number of incidents as to t merging clusters, $t_{C_i}$ is the centroid value of the response time of the *i*th cluster out and $w_i$ is the total number of incidents that belong to the *i*th merging cluster. The centroid value of the new merging cluster is equal to the mean value of the centroid of the margin of clusters. In Figure 9, the estimated response times in the left and right directions of the motorway are illustrated.

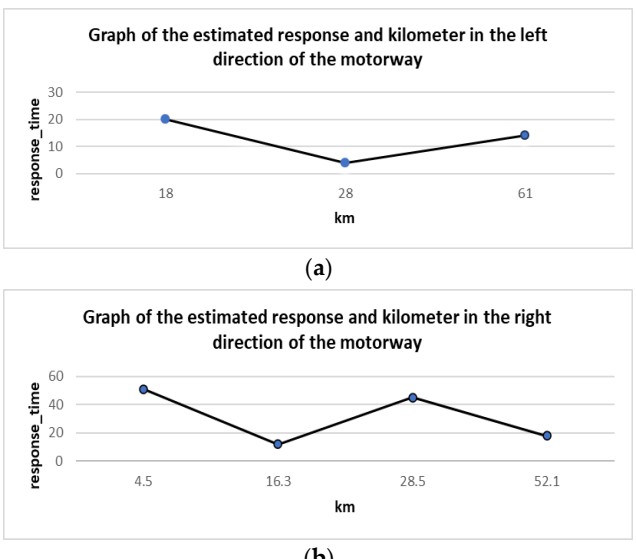

**Figure 9.** Output Graph of the estimated response time per field personnel cluster in (**a**) the left direction towards Igoumenitsa and (**b**) the right direction towards Ioannina.

Linear regression has been used to calculate the response time at the midpoints of the cluster centroids. Nevertheless, the implementation of the SVR machine of response time machine algorithm uses a Radial Basis Function (RBF) Gaussian kernel. The gamma parameter used for the left direction is equal to: $gamma = \frac{1}{N\sigma^2} = \frac{1}{34(Var(X))} = \frac{1}{34288.5} = 0.0001$, and for the right direction is equal to: $gamma = \frac{1}{N\sigma^2} = \frac{1}{34(Var(X))} = \frac{1}{27922.4} = 0.00004$. The constant parameters remained as they are in their default values., the estimated response time for the left and right directions of the SVR machine using 4 different kernels (a) linear, (b) RBF, (c) quadratic, (d) cubic, are illustrated in Figure 10.

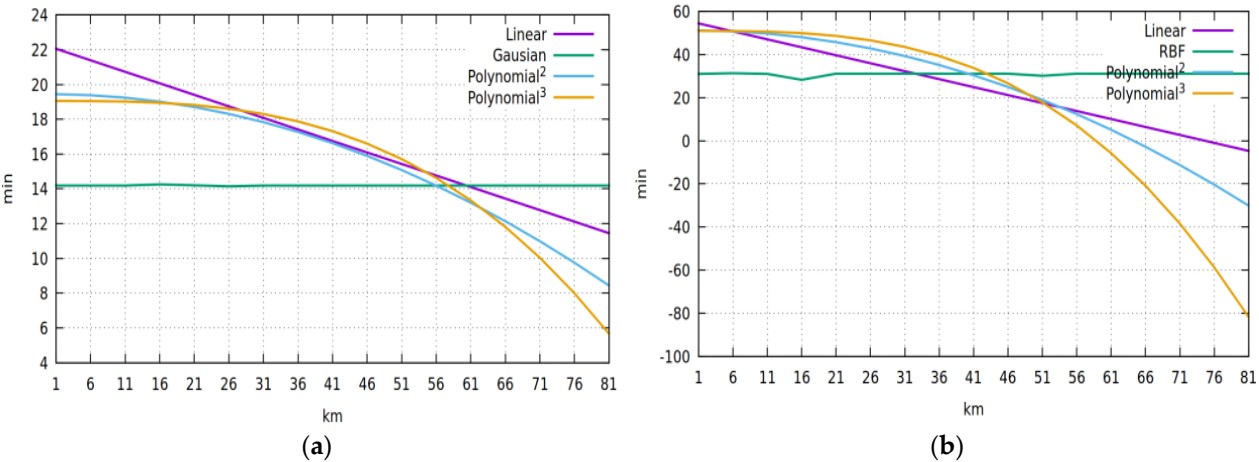

(a)  (b)

**Figure 10.** Plot of estimated time of field personnel in the (**a**) the left direction and (**b**) right directions on the motorway based on response time algorithm. SVR classifier response time prediction over distance.

SVR machine outputs show that in any case of the kernels' training data set is few. At least six months of data collecting would be adequate so the machine learning algorithm can work effectively. Moreover, the difference in the polynomial kernel of response time is obvious and clear. Comparison at Figure 10 shows that the difference in RBF's variation is proportional once the number of clusters increases. That means RBF is consistent when the data set is small and slightly changes-fluctuates occur when the data set is big, or the km section increases. In our case, the SVR linear kernel is the best optimal solution following the polynomial kernels to our problem for estimating response time. In the case of testing the whole length of the motorway, which is 658 km, the best optimal solution would be the RBF kernel.

*5.2. Completion Time Algorithm Evaluation*

On the data set of the RMS as they are inserted into the dataset and during the pilot application on the area of the 0–69 km of the motorway, from Ioannina to Igoumenitsa, the data filtering mechanism has been applied. The first step of the data filtering includes outliers' removal. The second one discretizes the values of the completion time and the last one discretizes the values of the completion time in the interval of [0, 1]. Incident descriptions are classified in descending order based on the frequency, and all of them are displayed through the Intelligent Agent (smart bot), so the RMS user can choose which one is the most suitable for the incident that occurred.

The goal is to create as many clusters as needed to reach the desired result of SSE to be less than 0.5. That is, to select the minimum sum of square distances that give a result of less than 0.5 or the sum of the square distance to the total number of clusters created is less than 0.5. The sums of the squared errors are normalized. In the given data set for the left lane, when the normalization of the data over the number of clusters equals 11 clusters, the total sum of squared distance is less than 0.5 as shown in Figure 11.

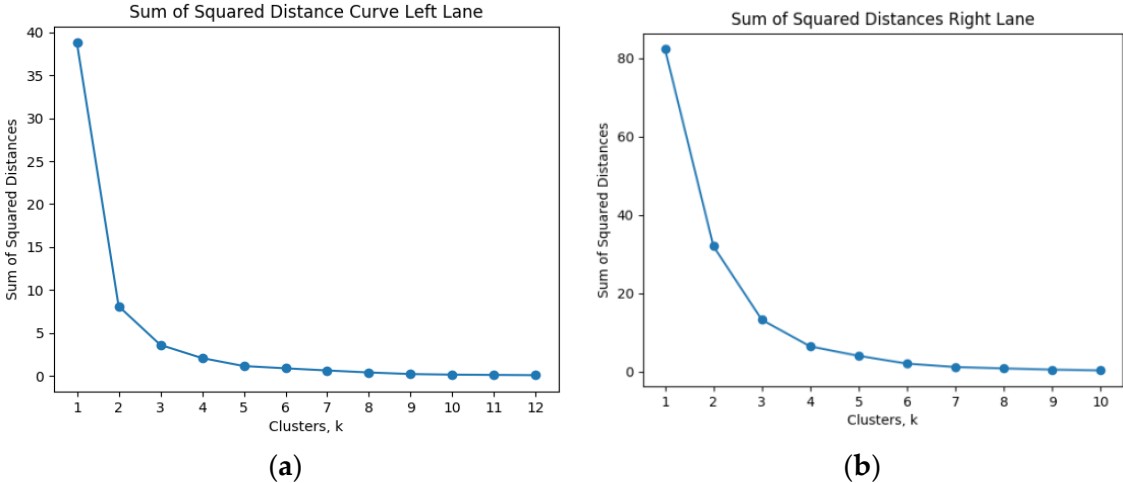

**Figure 11.** Sum of Squared Distances over k clusters for the (**a**) the left and (**b**) right lane.

As the *k*-Means algorithm does not give distinct values for the incident category, the square deviation between the incident category (1, 2, 3, 4) and the cluster centroids given by the algorithm was calculated. For the category of incidents, a rounding was performed based on the result given by the cluster rounding to the nearest integer. The function that rounds the centroids is given by Equation (12).

$$y = \begin{cases} floor(x)\,, & if\ x\ is\ not\ exceeded\ 1/2 \\ ceil(x)\,, & if\ x\ is\ exceeded\ 1/2 \end{cases} \tag{12}$$

The calculation of the square deviation between the value and the value given by the centroid of the *k*-Means algorithm shows the weight that needs to be multiplied by the Completion Time by *k*-Means to estimate the Completion Time. The weight of each cluster is described in Equation (13).

$$W_i = (y(centroid) - centroid)^2 \tag{13}$$

For the right lane, the normalization of the data over the number of clusters equals 9 clusters. The total sum of squared distance is less than 0.5, as shown in Figure 11. The result indicates the squared deviation between the rounded Incident Category and the value given by the *k*-Means algorithm (centroid). The aim is to calculate the difference, which can be achieved by the $1 - W_i$. The next step is to calculate the sum of the weights in a rounded form and divide it by the sum of the $1 - W_i$. The metric called aggregative metric is given by Equation (14).

$$Aggregative\ metric = \frac{\sum_{i=1}^{n}(1 - W_i * y(centroid_i))}{\sum_{i=1}^{n}(1 - W_i)} \tag{14}$$

Results have shown that for incident category 1 the required completion time is 108.15 min. For incident category 2 the needed completion time is 73.5 min. For incident category 3 the needed completion time is 4.1 min. For incident category 4 the required completion time is 27.3 min. Finally, the k-Nearest Neighbors algorithm is applied. The input to the algorithm is the set of data consisting of the Incident categories and Completion Time, and the output is defined as the Completion Time. Based on the Incident category (1, 2, 3, 4), that is to make the Completion Time predictable. The k-NN algorithm's model was examined for the Manhattan and Euclidean metrics. A random observation with a completion of 65 min, which belongs to Incident Category 2 was predicted for the left lane. Results show that for each of the metrics, the k-NN algorithm using Euclidean metric is 73%. Using the Manhattan metric, the k-NN algorithm is 90% accurate. For the right lane,

the input of the dataset consists of 6 incident categories (1, 2, 3, 4, 5, 9) to make Completion Time predictable. The k-NN algorithm is 54% accurate using Euclidean metric and 87% using the Manhattan. In both cases, the Manhattan metric outperforms the Euclidean one and is the one that is applied in the final k-NN model.

## 6. Conclusions

This paper presents a new monitoring and management motorway system called RMS. The proposed system is capable of automated real-time personnel detection and utilization per incident using the distance algorithm. Furthermore, the system includes an embedded evaluation mechanism for the incidents response and completion times, based on metrics and the proposed response and completion time algorithms. Distance, response, and completion time algorithms can be used to dynamically predict which team of personnel should respond to an incident on the motorway.

To automatically evaluate the team response and completion time of an incident and task, accordingly, concerning the motorway personnel team, is undoubtedly essential. Implementing new machine learning algorithms that include data mining processes using traditional machine learning techniques on selected best response scenarios, the incidents response time and tasks completion time can be significantly reduced. The proposed resources management system has been designed to automatically respond and mobilize personnel when motorway incidents offering dynamically estimated metrics and adaptive algorithms.

The system has been validated by the EGNATIA motorway end-users (Users Acceptance Tests—UATs) and validated its functionality and usefulness. Authors processed data given by the EGNATIA headquarters and the data from the experimental RMS system used for training and validation purposes [32]. The predictions taken from the RMS system have shown that in some cases, incidents require specific teams of personnel to be assigned the incident than other teams that are closer to the incident, even if the distances calculated from the proposed distance algorithm of the favorable by the algorithm team is less.

The experimental results have shown that almost 30% of the incidents in both lanes are equally distributed, and 70% occur by cars moving from the Eastern part of Greece to the Western. The experimentation results of applying the response time algorithm have shown that the best optimal case for getting accurate response time predictions from the data is to cluster the motorway area into areas where teams are moving along to distances up to 200–250 km and the most accurate SVM kernel for such predictions is the linear SVR kernel. However, if the teams can move and offer services in distances more than 80–120 km, the RBF kernel can derive better response time results are the SVR polynomial kernels.

Experimentation of applying the completion time algorithm using only incident categories and their descriptions have shown that each of the incident categories examined to give the appropriate estimated completion time derived from the RMS collected data. The two cases tested in the k-NN algorithm are the Euclidean and Manhattan metric. The Manhattan metric outperformed the Euclidean. For the left lane, the difference in accuracy rate between the two metrics is 17%, and for the right one, the difference between the two metrics is 33%. In this case, the Manhattan metric gives the best predictions.

**Author Contributions:** Conceptualization, S.K. and C.A.; methodology, S.K.; software, S.K. and C.A.; writing—original draft preparation, C.A.; writing—review and editing, S.K.; visualization, S.K. and C.A.; supervision, S.K.; funding acquisition, S.K. All authors have read and agreed to the published version of the manuscript.

**Funding:** This research has been co-financed by the European Union and Greek national funds through the Operational Program Competitiveness, Entrepreneurship, and Innovation, under the call RESEARCH—CREATE—INNOVATE (project code: T1EDK-02374).

**Institutional Review Board Statement:** Not applicable.

**Informed Consent Statement:** Not applicable.

**Data Availability Statement:** 3rd Party Data. Restrictions apply to the availability of these data. Data was obtained from EGNATIA ODOS SA and are available at: https://hmrt.math.uoi.gr (accessed on 10 May 2011) with the permission of EGNATIA ODOS SA.

**Acknowledgments:** Project partners: Sector of Industrial Management and Operational Research of the National Technical University of Athens, EGNATIA Motorway SA, TEKMON P.C., Department of Mathematics of the University of Ioannina, National Centre of Scientific Research "DEMOKRITOS". The authors would also like to acknowledge A. Saramourtsis, A. Tsantsanoglou, and G. Godevenos from EGNATIA ODOS SA for their continuous support towards this research providing ideas for the under-development RMS.

**Conflicts of Interest:** The authors declare no conflict of interest.

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
