# Peer review of "Proposed Management System and Response Estimation Algorithm for Motorway Incidents"

_energies, doi:10.3390/en14102736_

Round 1
Reviewer 1 Report
This paper presents a new holistic and unified management and response system called Resources Management System (RMS). This system is used by the EGNATIA ODOS motorway in Greece
Can RMS only used in motorways? or it can be used in normal streets?
What is the advantages and disadvantage of the RMS?
I would recommend to compare with state-of-the-art approach.
Author Response
Comments from Reviewer 1:
Comment 1:
This paper presents a new holistic and unified management and response system called Resources Management System (RMS). This system is used by the EGNATIA ODOS motorway in Greece.
Can RMS only used in motorways? or it can be used in normal streets?
Response 1:
The proposed RMS has been implemented and tested for motorways use. It takes into consideration the motorways’ tunnels; the latitude and longitude of the kilometer position marks each motorway. However, Its core functionality and algorithms for incidents response that take into account response time and completion time (not the use of the distance algorithm) it can also be used in streets as well.
Comment 2:
What are the advantages and disadvantage of the RMS?
Να απαντήσουμε σε αυτÏŒ
Response 2:
Advantages of the proposed RMS has been added at section 3, paragraph 2 justifying the RMS system’s holistic approach (RMS services and mapping with facility management, incident response and sensory equipment measurements acquisition).
The proposed RMS helps tracking, monitoring, and sending notifications to the users who are on the motorway to notify them directly about incidents instead of calling or notifying them using other communication means. Regarding facility management, the use of the mobile phone application where personnel can fill application forms in the field and event add incidents in the system is a novel approach. Moreover, RMS reduces the time that people would spend to fill out reports. Finally, the use of a unified way – direct database storage of sensory measurements, using a scalable open-source sensory distribution system approach is also a significant advantage over the existing closed source and custom protocol-based SCADA systems.
Comment 3:
I would recommend to compare with state-of-the-art approach.
Response 3:
Appropriate Table 1 has been added at section 3 for state-of-the-art comparison of the proposed RMS system with existing systems.
Reviewer 2 Report
This paper proposed a Resources Management System. The RMS provides real-time resources tracking, personnel utilization algorithms, and data mining capabilities towards incident confrontation. And, the RMS includes a machine learning methodology and smart agents (bots) to estimate and evaluate the response and completion time of incidents, based on previous cases. It is claimed to be a new holistic and unified management and response system, but which part(s) are new should be identified clearer in the Abstract and Introduction. The introduction started from introducing machine learning, which gave a wrong expression that the contribution of the paper was to develop new machine learning techniques, whereas the ML techniques used in the project were rather standard, e.g. k-means, SVR and kNN. Instead, the introduction should start with the road transportation and the background/context of the project, whereas the ML techniques could be introduced or discussed later for future improvement. In additional to the ML techniques, optimisation methods should also be discussed, as a few parts in the architecture involve finding the optimal parameters/hyperparameters, referring to, grey-box modelling of the swirl characteristics in gas turbine combustion system, measurement. Various results and figures should be justified or discussed. E.g., the fitting of Gaussian mixture model, the choice of kernels, the choice of the number of clusters, referring to, selecting optimal features for cross-fleet analysis and fault diagnosis of industrial gas turbines, asme turbo expo. Explanations of the EM algorithm and other standard techniques could be reduced. Quality of figures should be improved. Figure 1 which is the essence of the proposed RMS could be made more representative. Figure 3, Figure 5 and Figure 6 should be improved with appropriate flowcharts. Format of the result figures should be unified. Formatting of equations and references in text should be checked also.
Author Response
Comment 1:
This paper proposed a Resources Management System. The RMS provides real-time resources tracking, personnel utilization algorithms, and data mining capabilities towards incident confrontation. And, the RMS includes a machine learning methodology and smart agents (bots) to estimate and evaluate the response and completion time of incidents, based on previous cases. It is claimed to be a new holistic and unified management and response system,
but which part(s) are new should be identified clearer in the Abstract and Introduction.
Response 1:
A clearer and briefer description of the new parts of the RMS capabilities have been added in the Abstract and Introduction.
Comment 2:
Started from introducing machine learning, which gave a wrong expression that the contribution of the paper was to develop new machine learning techniques, whereas the ML techniques used in the project were rather standard, e.g. k-means, SVR and kNN. Instead, the introduction should start with the road transportation and the background/context of the project, whereas the ML techniques could be introduced or discussed later for future improvement.
Response 2:
The contribution of this paper is to develop smart data mining and machine learning algorithms for the process of automated incidents response confrontation and tasks management. Road transportation, road tunnel safety, resource management systems and the contribution of the paper have decently introduced in the introduction part.
Comment 3:
In additional to the ML techniques, optimisation methods should also be discussed, as a few parts in the architecture involve finding the optimal parameters/hyperparameters, referring to, grey-box modelling of the swirl characteristics in gas turbine combustion system, measurement. Various results and figures should be justified or discussed. E.g., the fitting of Gaussian mixture model, the choice of kernels, the choice of the number of clusters, referring to, selecting optimal features for cross-fleet analysis and fault diagnosis of industrial gas turbines, asme turbo expo.
Response 3:
This paper presents the authors’ proposition of ML algorithms implementation to an holistic RMS system. The algorithms functionality has been validated. The process of algorithms optimization is part of an on-going work of evaluating the algorithms at the EGNATIA motorway. The evaluation results will be part of a future work.
Comment 4:
Explanations of the EM algorithm and other standard techniques could be reduced.
Response 4:
The main steps of the EM algorithm have been removed.
Comment 5:
Quality of figures should be improved. Figure 1 which is the essence of the proposed RMS could be made more representative. --- use 300dpi
Response 5:
The quality of figures has been improved to 300dpi. Figure 1 which essentially is the proposed resources management system has been made more representative.
Comment 6:
Figure 3, Figure 5 and Figure 6 should be improved with appropriate flowcharts. Format of the result figures should be unified. Formatting of equations and references in text should be checked also.
Response 6:
Figures 3,5 and 6 have been improved with appropriate flowcharts. The format of the result figures cannot be unified since each subfigure of a figure represents a different result statement. The format of equations and references in text have been corrected.
Reviewer 3 Report
Review of paper 1190869
Making the Proposed management system and response estimation algorithm for motorway incidents, in the present age, when monitoring incidents and collecting information related to accidents are vital processes, is a quite important effort for the research staff. The submitted paper proposes a Resources Management System for contemporary motorways and corresponding tunnels. There are multiple aspects which are considered in this case, such as tunnel temperature, humidity, and the speed of the vehicles. The proposed solution is handling the automated real-time personnel detection and utilization per incident with the use of the distance algorithm.
Introduction |
|
Does the introduction provide sufficient background information for readers not in the immediate field to understand the problem/hypotheses? |
The initial part of the paper makes a balanced and generalized foundation of the topic which shortly gives to any interested reader an overview of the problem regarding Proposed Resources Management System Architecture. |
Are the reasons for performing the study clearly defined? |
Yes |
Are the study objectives clearly defined? |
The objectives are well defined in the Introduction. |
2. literatures Review and Model Construct |
|
Is the literature cited balanced or are there important studies not cited, or other studies disproportionately cited? |
The literature cited in this paper is quite relevant to the research. |
Please identify statements that are missing any citations, or that have an insufficient number of citations, given the strength of the claim made. |
- |
3. Methodology and Data |
|
Are the methodology and data used appropriate to the purpose of the research? |
Yes |
Is sufficient information provided for a capable researcher to reproduce the experiments described? |
The authors should probably provide more information about the local conditions for Road Management System (RMS). |
Are any additional experiments required to validate the results of those that were performed? |
I think some experiments are necessary to validate the results presented here, because the results themselves are also important; as well as the technique used to obtain these results is important. |
Are there any additional experiments that would greatly improve the quality of this paper? |
Yes |
Are appropriate references cited where previously established methods are used? |
Yes |
4. Results |
|
Are the results clearly explained and presented in an appropriate format? |
Yes |
Do the figures and tables show essential data or are there any that could easily be summarized in the text? |
Yes. |
Are any of the data duplicated in the graphics and/or text? |
No. |
Are the figures and tables easy to interpret? |
It may be improved. |
Are there any additional graphics that would add clarity to the text? |
Yes. |
Have appropriate statistical methods been used to test the significance of the results? |
Yes. |
5. Conclusions and Implications |
|
Are all possible interpretations of the data considered or are there alternative hypotheses that are consistent with the available data? |
It may be improved. |
Are the findings properly described in the context of the published literature? |
Yes |
Are the limitations of the study discussed? If not, what are the major limitations that should be discussed? |
May be further defined. |
Are the conclusions of the study supported by appropriate evidence or are the claims exaggerated? |
Conclusions of the study are supported by the results. |
Significance and Novelty |
|
Are the claims in the paper sufficiently novel to warrant publication? |
Yes. |
Does the study represent a conceptual advance over previously published work? |
Yes. |
Journal Selection |
|
Is the target journal (if known) appropriate? If not, why not? |
Yes |
What is the likely target audience of this paper? |
Road traffic engineers. Road managers. Road traffic research units.. |
Minor comments |
|
Please refer to the comments in the edited manuscript file for minor comments. |
Please make shorter sentences. The longer ones are difficult to be understood. |
Major comments |
|
To publish this paper in your target journal, the following revisions are strongly advised: |
- |
There are multiple [Error! Reference source not found.]
The normal distributions of the data set as well as the calculation of the likelihood logarithm of the chosen distribution(s) as to the sum of all the probabilities of the data set points (A) extract are described in Error! Reference source not found.:
Figure 10. Plot of estimated time of field personnel in the (a) the left
Author Response
Comment 1:
2.literatures Review and Model Construct
Is the literature cited balanced or are there important studies not cited, or other studies disproportionately cited? Please identify statements that are missing any citations, or that have an insufficient number of citations, given the strength of the claim made.
Response 1:
New citations have been added regarding the unified sensory data collection and data concentration and Bluetooth sensory detectors implemented by the authors at the motorway tunnel exits.
Comment 2:
Are all possible interpretations of the data considered or are there alternative hypotheses that are consistent with the available data?
Response 2:
This paper presents a holistic system approach for motorways which has been validated for its algorithms’ response. Further system evaluation is an ongoing future work.
Comment 3:
Are the limitations of the study discussed? If not, what are the major limitations that should be discussed?
Response 3:
The main limitation of the study includes a thorough evaluation of the proposed algorithms and bots supporting those algorithms as well as scalability of the proposed sensory management sub-system architecture. Such limitations are considered an ongoing future work by the authors.
Comment 4:
Journal Selection
Is the target journal (if known) appropriate? If not, why not?
Response 4:
This Energies special session is about industry 4.0 advances with the use of unified, scalable and collective processes combining also the use of smart data derived decisions.
Comment 5:
What is the likely target audience of this paper?
Response 5:
The target audience of this paper is motorway operators that seek for an holistic motorway monitoring approach that combines 1) Facility management/tasks assignment -personnel tracking and evaluation, 2) Incidents response and automated assignment of resources per incident based on distance or response time and 3) a unified approach for the collection of sensory measurements spread out on motorways.
Comment 6:
Please make shorter sentences. The longer ones are difficult to be understood.
Response 6:
The manuscript has been reviewed, checked for typo errors and syntax errors and long sentences have been shortened where possible.
Comment 7:
There are multiple [Error! Reference source not found.]
The normal distributions of the data set as well as the calculation of the likelihood logarithm of the chosen distribution(s) as to the sum of all the probabilities of the data set points (A) extract are described in Error! Reference source not found.:
Response 7:
Manuscript references have been thoroughly checked and bibtex reference errors have been corrected.
Comment 8:
Figure 10. Plot of estimated time of field personnel in the (a) the left
Response 8:
Figure 10 captions have been fixed.
Round 2
Reviewer 2 Report
Response was brief, and only part of the comments were addressed - the flowcharts could be made more concise and representative, while the quality of the figures for results could be improved.
Author Response
Changes and corrections have been made. Minor spelling errors have been fixed. in terms of English language and style.
